# Multi-objective optimal decision for orderly power utilization based on improved ε-constraint method in active distribution networks

Xin Wen[1], Hui Li[1], Xiaoqiang Wu[1], Yiwei Li[1], Liu Siliang[2]*, Guohua Huang[2]

1 Guangzhou Power Supply Bureau, Guangdong Power Grid Co., Ltd., Guangzhou, China, 2 Guangzhou Power Electrical Engineering Technology Co., Ltd., Guangzhou, China

* 523110355@qq.com

**Data Availability Statement:** All relevant data are within the paper and its Supporting information.

**Funding:** This research received funding Science and Technology Project of China Southern Power

## Abstract

With the increasing demand for electricity load in China, orderly power utilization are important measures to alleviate electricity shortages during peak periods. This article establishes a multi-objective optimization model for orderly power utilization in active distribution networks is established, with the optimization objectives of minimizing the total operation cost, minimizing the cost for users, and minimizing the load fluctuation of the system. This model contains a large number of integer variables and nonlinear constraints, which is difficult to solve. To reduce computation time, convex relaxation techniques are adopted to transform the original model into a mixed-integer second-order cone programming (MISOCP) model, which has lower computational complexity. Furthermore, The improved ε-constraint method is proposed to solve the model, which can directly and quickly find the compromise optimal solution of the multi-objective problem. By using simplex search algorithm, the proposed method dose not need to traverse all grid points, which can significantly reduce computation time. Finally, case study on the the IEEE-33 bus distribution network demonstrate the effectiveness of the proposed method.

## 1 Introduction

With China's economic development, the electricity load is growing rapidly [1]. When power supply is insufficient, the impact of power shortage on national economic life can be minimized through the implementation of orderly power utilization and the adoption of administrative measures, economic means or technical methods to maintain the balance of power supply and utilization [2]. Demand response [3] can be used as an auxiliary means to promote the balance between power supply and demand, but it cannot ensure the response to the power load gap during peak periods. Therefore, it is of great practical significance and application value to study the decision-making method of orderly power utilization.

At present, there have been numerous scholars to carry out research on the decision-making of orderly power utilization. Peng et al. [4] establish the objective function by summing up

Grid Corporation (GZHKJXM20210056 (080036KK52210003)).The funders play an important role in study design and decision to publish.

**Competing interests:** The authors have declared that no competing interests exist.

the compensation cost for orderly power utilization methods and the cost associated with network loss penalties. By incorporating the network loss penalty term, they prioritize the utilization of power sources during the decision-making process for orderly power utilization. Zhang et al. [5] formulate a decision-making model for orderly power utilization, which coordinates both weekly and daily time scales. The objective of this model is to minimize the combined cost of peak avoidance and the penalty associated with fluctuations in peak and valley loads. The above literature are all single objective optimization model, both of them only consider the control operation cost of orderly power utilization, but not the impact of orderly power utilization measures on users. Contrasting the single objective optimization model, the multi-objective optimization model offers numerous non-dominated solutions, also known as the Pareto optimal solutions. The decision-maker need carefully choose a compromise optimal solution from among these Pareto optimal solutions. The approaches to solving the multi-objective optimization model can be categorized into heuristic optimization methods [6, 7] and traditional optimization techniques [8, 9]. Heuristic optimization methods offer the flexibility of directly solving the model without the limitations on its characteristics. Nevertheless, their computational inefficiency renders them impractical for real-world distribution networks. As for the traditional optimization methods, they mainly include the weighted sum method and the $\varepsilon$-constraint method. When using traditional optimization methods, there are usually restrictions on the type of model, and it is necessary to perform convex relaxation on the model. Wang et al. [8] propose a multi-objective optimization decision model for orderly power utilization and adopt the weighted sum method to solve the proposed model. The model aims at taking maximizing the grid load rate and maximizing the energy efficiency of power utilization. Paterakis et al. [9] adopt the $\varepsilon$-constraint method to solve a multi-objective optimization model, which aims to determine the optimal energy and reserve volumes considering the joint energy and reserve day-ahead market. The $\varepsilon$-constraint method divides the range of values of $\varepsilon_i$ into $q$-equal segments and generates distributed grid points. By traversing all grid points, the set of Pareto optimal solutions can be obtained. As the value of $q$ increases, there will be more grid points, which will seriously affects the computation efficiency.

In summary, this paper makes two contribution: 1) A multi-objective optimization model for orderly power utilization in active distribution networks with the optimization objectives of minimizing the total operation cost, minimizing the cost for users, and minimizing the load fluctuation of the system. For the nonlinear constraints in the model, convex relaxation techniques are adopted to transform the original model into a mixed-integer second-order cone programming (MISOCP) model, which has lower computational complexity. 2) The improved $\varepsilon$-constraint method is proposed to solve the model, which can directly and quickly find the compromise optimal solution of the multi-objective problem. Compared to the traditional $\varepsilon$-constraint method, the proposed method dose not need to traverse all grid points by using simplex search method, which can significantly reduce computation time.

## 2 Problem formulation

Means of orderly electricity use include peak transferring, peak shifting, peak clipping, and power cut. Peak transferring refers to transferring the total power load of users during a period by advancing or lagging time.The starting time of peak transferring is unset, and the advancing or lagging time should be less than the maximum allowed time of peak transferring. Peak shifting refers to shifting part of the power load of users from the peak period to the valley period, and the time of peak period and valley period is set. Peak clipping refers to restricting part of the user's load during the peak period. Power cut forces the power supply to be stopped when the other means are exhausted and the generation and use of electricity are still unbalanced.

Due to the serious impact of power cut on national economy and people's well being, this paper does not consider this measure. Considering the impact on users' electricity consumption behavior, peak transferring only requires appropriate adjustment of the time of electricity consumption, without reducing electricity consumption. Peak shifting requires adjustment of the power utilization plan within a day, without reducing electricity consumption. But peak clipping requires reduction in electricity consumption.

In summary, this paper considers the order of invocation of the decision-making tools for orderly power use as: peak transferring, peak shifting, and peak clipping. In this paper, the graded peak transferring and graded peak shifting are considered. Each user can participate in different grades of peak shifting, and graded peak clipping is an integer multiple of the amount of peak clipping in a single grade. At the same time, considering the distributed renewable energy field stations and energy storage power stations in the active distribution network, the system load transfer is coordinated and the wind and light resources are better absorbed by reasonably formulating the charging and discharging plan of the energy storage power stations.

## 2.1 Objective function

**2.1.1 Minimizing total operation cost.** The total operating cost of the distribution network includes the compensation cost for orderly power utilization, the renewable energy curtailment penalty cost and the operating cost of the energy storage. Among them, the compensation cost for orderly power utilization can be further divided into the compensation cost for peak transferring, peak shifting, and peak clipping, as shown below:

$$f_1 = \sum_{i=1}^{N}(C_{trans,i} + C_{shift,i} + C_{clip,i}) + \sum_{t=1}^{T}(C_{RES,t} + C_{ES,t}) \tag{1}$$

Where, the subscript $i$ denotes the user number connected to bus $i$. $N$ and $T$ are the total number of system buses and time periods. $C_{trans,i}$, $C_{shift,i}$ and $C_{clip,i}$ are the peak transferring, peak shifting and peak clipping compensation cost for user $i$, respectively. $C_{RES,t}$ is the penalty cost for system renewable energy curtailment at time $t$, and $C_{ES,t}$ is the operation cost of the energy storage station at time $t$.

The specific formula for each item in Eq (1) is given below:

$$C_{trans,i} = \sum_{h=-h_{max,i}}^{h_{max,i}} \lambda_{trans} u_{i,h}^{trans} |h| \tag{2}$$

Where, $\lambda_{trans}$ is the compensation cost coefficient of peak transferring per unit time period, in yuan/h. $u_{i,h}^{trans}$ is a 0–1 variable, and when it takes the value of 1, it means that user $i$ peak transferring $h$ hours. If the values of $h$ is positive, it means that the power utilization time is advanced. If not, it means that the power utilization time is lagging. $h_{max,i}$ is the maximum allowed time of peak transferring for user $i$.

$$C_{shift,i} = \sum_{\beta=1}^{S_i} \left( \lambda_{shift} u_{i,\beta}^{shift} \frac{1}{2} \sum_{t=1}^{T} \left| \Delta PL_{i,\beta,t}^{shift} \right| \right) \tag{3}$$

Where, $\lambda_{shift}$ is the compensation cost coefficient of peak shifting per unit time period, in yuan/MW·h. $S_i$ is the total number of peak shifting that the $i$-th user can participate in. $u_{shift\ i,\beta}$ is a 0–1 variable, and when it takes the value of 1, it means that the $i$-th user participates in the $\beta$-th peak shifting. $\Delta PL_{i,\beta,t}^{shift}$ is the load correction corresponding to the $\beta$-th peak shift of user $i$

at time $t$. Since the absolute value of positive and negative corrections of peak and valley shifts in a day are equal, the actual peak shifting is the sum of the absolute value of load corrections in each time period divided by two.

$$C_{clip,i} = \lambda_{clip} C_i \sum_{t=1}^{T} \Delta P_{i,t}^{clip} \tag{4}$$

Where, $\lambda_{clip}$ is the compensation coefficient of peak clipping per unit time period, in yuan/MW·h. $C_i$ is the number of stages in which user $i$ participates in peak clipping, and $\Delta P_{clip\ i,t}$ is the amount of single-stage load modification in which user $i$ participates in peak clipping at time $t$.

$$C_{RES,t} = \lambda_{RES} \left( \sum_{i=1}^{N_s}(P_{sf,i,t} - P_{s,i,t}) + \sum_{i=1}^{N_w}(P_{wf,i,t} - P_{w,i,t}) \right) \tag{5}$$

Where, $\lambda_{RES}$ is the penalty cost coefficient of renewable energy curtailment, in yuan/MW·h. $N_s$ and $N_w$ are the total number of photovoltaic power stations and wind farms, respectively. $P_{sf,i,t}$ and $P_{wf,i,t}$ are the predicted active output of photovoltaic power stations and wind farms at time $t$, respectively. And $P_{s,i,t}$ and $P_{w,i,t}$ are the scheduled output of photovoltaic power stations and wind farms at time $t$, respectively. The predicted output is always larger than the scheduled output, and the difference between them is the amount of renewable energy curtailment.

$$C_{ES,t} = \lambda_{ES}(P_{c,i,t} + P_{d,i,t}) \tag{6}$$

Where, $\lambda_{ES}$ is the charging and discharging operation cost coefficient of the energy storage station, in yuan/MW·h. $P_{c,i,t}$ and $P_{d,i,t}$ are the charging active power and discharging active power of the energy storage plant at moment t, respectively.

**2.1.2 Minimizing the cost for users.** The cost of electricity to the user is the sum of the cost of electricity purchased by all users of the system, as shown in the following equation:

$$f_2 = \sum_{t=1}^{T} \sum_{i=1}^{N} c_{grid,t} PL_{i,t} \tag{7}$$

Where, $c_{grid,t}$ is the time-of-day tariff. $PL_{i,t}$ is the actual load value of user $i$ in time period $t$, as shown in the following equation:

$$\begin{cases} PL_{i,t} = PL_{i,t,0} - \Delta PL_{i,t}^{trans} - \Delta PL_{i,t}^{shift} - \Delta PL_{i,t}^{clip} \\ \Delta PL_{i,t}^{trans} = \sum_{h=-h_{max,i}}^{h_{max,i}} u_{i,h}^{trans}(PL_{i,t,0} - PL_{i,t+h,0}) \\ \Delta PL_{i,t}^{shift} = \sum_{\beta=1}^{S_i} u_{i,\beta}^{shift} \Delta PL_{i,\beta,t}^{shift} \\ \Delta PL_{i,t}^{clip} = C_i \Delta P_{i,t}^{clip} \end{cases} \tag{8}$$

Where, $PL_{i,t,0}$ is the initial load value of user $i$ at time $t$. $\Delta PL_{i,t}^{trans}$, $\Delta PL_{i,t}^{shift}$ and $\Delta PL_{i,t}^{cilp}$ are the load corrections for peak transferring, peak shifting and peak clipping for user $i$ at time $t$, respectively.

**2.1.3 Minimizing system load standard deviation.** For the load fluctuation of the distribution network, the standard deviation of load active power can be expressed. The system load standard deviation reflects the smoothness of the load active power [10]. In order to alleviate

the system peak load demand and reduce the peak-valley difference, the system load standard deviation should be minimized, and its calculation equation is shown below:

$$f_3 = \sqrt{\frac{1}{N}\sum_{t=1}^{T}\left(\sum_{i=1}^{N}PL_{i,t} - \frac{1}{T}\sum_{t=1}^{T}\sum_{i=1}^{N}PL_{i,t}\right)^2} \qquad (9)$$

## 2.2 Objective function

**2.2.1 Power flow equations.** The AC current equation is used to describe the power balance constraints at each bus in the distribution network as shown in the following equation:

$$\begin{cases} P_{RES,t} - PL_{i,t} + P_{ES,t} = V_{i,t}\sum_{j=1}^{N}V_{j,t}(G_{ij}\cos\theta_{ij,t} + B_{ij}\sin\theta_{ij,t}) \\ \\ Q_{RES,t} - QL_{i,t} = V_{i,t}\sum_{j=1}^{N}V_{j,t}(G_{ij}\sin\theta_{ij,t} - B_{ij}\cos\theta_{ij,t}) \end{cases} \qquad (10)$$

Where, $P_{RES,t}$ and $Q_{RES,t}$ are the active and reactive power of the renewable energy field station at time $t$, respectively. $Q_{Li,t}$ is the reactive load of bus i at time $t$, $P_{ES,t}$ is the active power of the energy storage power station at time $t$. $V_{i,t}$ is the voltage amplitude of bus $i$ at time $t$. $G_{ij}$ and $B_{ij}$ are the mutual conductance/susceptance between buses $i$ and $j$, respectively. $\theta_{ij,t}$ is the voltage phase difference between buses $i$ and $j$ at time $t$. If bus $i$ is connected to a renewable energy station or an energy storage station, the formulas for $P_{RES,t}$, $Q_{RES,t}$ and $P_{ES,t}$ in Eq (10) are shown in Eq (11). Otherwise, they take the value of 0.

$$\begin{cases} P_{RES,t} = P_{s,i,t} + P_{w,i,t} \\ Q_{RES,t} = Q_{s,i,t} + Q_{w,i,t} \\ P_{ES,t} = P_{d,i,t} - P_{c,i,t} \end{cases} \qquad (11)$$

Where, $Q_{s,i,t}$ and $Q_{w,i,t}$ are the scheduled reactive power output of the PV plant and wind farm at time t, respectively.

**2.2.2 Constraints of renewable energy station operation.** The operation constraints of renewable energy station are shown in Eqs (12) and (13). Assume that the wind farm operate in the constant power factor mode.

$$\begin{cases} 0 \leq P_{s,i,t} \leq P_{sf,i,t} \\ P_{s,i,t}^2 + Q_{s,i,t}^2 \leq S_{s,i}^2 \\ -P_{s,i,t}\tan\varphi_{s,i} \leq Q_{s,i,t} \leq P_{s,i,t}\tan\varphi_{s,i} \end{cases} \qquad (12)$$

$$\begin{cases} 0 \leq P_{w,i,t} \leq P_{wf,i,t} \\ Q_{w,i,t} = P_{s,i,t}\tan\varphi_{w,i} \end{cases} \qquad (13)$$

Where, $S_{s,i}$ is the apparent power of the photovoltaic power station. $\varphi_{s,i}$ and $\varphi_{w,i}$ are the power factor angles of photovoltaic power station and wind farm, respectively.

**2.2.3 Constraints of energy storage operation.** When the energy storage power station is in operation, the charging and discharging power must be smaller than its maximum operating power. Also, charging and discharging cannot be carried out at the same time, and the

remaining power should be within the specified power range.

$$\begin{cases} 0 \leq P_{c,i,t} \leq P_{ES,\max} \\ 0 \leq P_{d,i,t} \leq P_{ES,\max} \\ P_{c,i,t} \times P_{d,i,t} = 0 \\ E_{i,t} = E_{i,t-1} + \eta_c \times P_{c,i,t} - P_{d,i,t}/\eta_d \\ SOC_{\min} \leq E_{i,t}/E_{\max} \leq SOC_{\max} \end{cases} \tag{14}$$

Where, $P_{ES,\max}$ is the maximum operating power of the energy storage plant. $E_{i,t}$ is the remaining power of the energy storage plant at time $t$. $\eta_c$ and $\eta_d$ are the charging and discharging efficiencies of the energy storage plant, respectively. $E_{\max}$ is the maximum storage power of the energy storage plant. $SOC_{\min}$ and $SOC_{\max}$ are the minimum and maximum values of the charging state, respectively.

**2.2.4 Constraints of peak transferring.**   For all users, the number of hours participating in peak transferring in a day is unique. In particular, when $u_{i,0}^{trans} = 1$, it means that the user has 0 hour of time peak transferring.

$$\sum_{h=-h_{\max,i}}^{h_{\max,i}} u_{i,h}^{trans} = 1 \tag{15}$$

**2.2.5 Constraints of peak shifting.**

$$u_{i,0}^{shift} \leq \sum_{\beta=1}^{S_i} u_{i,\beta}^{shift} \leq S_i u_{i,0}^{shift} \tag{16}$$

Where, $u_{shift\ i,0}$ is a 0–1 variable that, when taking the value of 1, indicates that user $i$ participates in peak shifting and when taking the value of 0, indicates that user $i$ does not participate in peak shifting.

**2.2.6 Constraints of peak clipping.**

$$u_{i,0}^{clip} \leq C_i \leq u_{i,0}^{clip} C_{i,\max} \tag{17}$$

Where, $C_{i,\max}$ is the maximum level of peak clipping. $u_{clip\ i,0}$ is a 0–1 variable that, when taking the value of 1, indicates that user $i$ participates in peak clipping and when taking the value of 0, indicates that user $i$ does not participate in peak clipping.

**2.2.7 Constraints of participation modalities.**   For any user $i$, it can only participate in one of the peak transferring, peak shifting, and peak clipping methods of orderly power usage in a single day.

$$(1 - u_{i,0}^{trans}) + u_{i,0}^{shift} + u_{i,0}^{clip} \leq 1 \tag{18}$$

## 2.3 Convex relaxation of the model

The above equations constitute a multi-objective optimization model for orderly power utilization in active distribution networks. The model contains a large number of 0–1 variables, and there are nonlinear terms in Eq (10) and the third constraint in Eq (14). Therefore, the model is a mixed integer nonlinear programming, which is complex and takes a long time to solve. In

order to improve the computer efficiency, the non-convex constraints in the model are subjected to convex relaxation.

For Eq (10), the second order cone convex relaxation method [11] is used. Introduce variables $R_{ij,t} = V_{i,t}V_{j,t}\cos\theta_{ij,t}$, $T_{ij,t} = V_{i,t}V_{j,t}\sin\theta_{ij,t}$, $U_{i,t} = V2_{i,t}$, and then Eq (10) can be transformed into Eq (19).

$$
\begin{cases}
P_{RES,t} - PL_{i,t} + P_{ES,t} = U_{i,t}G_{ii} + \sum_{j=1,j\neq i}^{n}(R_{ij,t}G_{ij} + T_{ij,t}B_{ij}) \\[2mm]
Q_{RES,t} - QL_{i,t} = -U_{i,t}B_{ii} + \sum_{j=1,j\neq i}^{n}(T_{ij,t}G_{ij} - R_{ij,t}B_{ij}) \\[2mm]
\| \ 2R_{ij,t}; \quad 2T_{ij,t}; \quad U_{i,t} - U_{j,t} \ \|_2 - (U_{i,t} + U_{j,t}) \leq 0 \\[2mm]
R_{ij,t} - R_{ji,t} = 0, T_{ij,t} + T_{ji,t} = 0 \\[2mm]
\sum_{ij\in\Omega_k}[\tan^{-1}(T^0_{ij,t}/R^0_{ij,t}) + \dfrac{(R^0_{ij,t}T_{ij,t} - T^0_{ij,t}R_{ij,t})}{(R^0_{ij,t})^2 + (T^0_{ij,t})^2}] = 0
\end{cases}
\tag{19}
$$

For the third constraint in Eq (14), it can be linearized by using the big M method [12] and transformed into Eq (20) as follows:

$$
\begin{cases}
0 \leq P_{c,i,t} \leq M \cdot \zeta_{i,t} \\
0 \leq P_{d,m,t} \leq M \cdot (1 - \zeta_{i,t})
\end{cases}
\tag{20}
$$

Where, M is a large constant and $\zeta_{i,t}$ are auxiliary binary variables introduced by the big M method.

After performing the convex relaxation process, the model is transformed into a mixed-integer second-order cone programming. For descriptive convenience, the model is expressed in the compact form shown in Eq (21).

$$
\min\{f_1(\boldsymbol{x}), f_2(\boldsymbol{x}), f_3(\boldsymbol{x})\}
$$
$$
\text{s.t.}\begin{cases}
\boldsymbol{g}(\boldsymbol{x}) = 0 \\
\boldsymbol{h}(\boldsymbol{x}) \leq 0
\end{cases}
\tag{21}
$$

Where, $\boldsymbol{x}$ denotes all the variables in the model. $\boldsymbol{g}(\boldsymbol{x})$ and $\boldsymbol{h}(\boldsymbol{x})$ denote the equation constraints and inequality constraints in the model, respectively.

## 3 Model solution

The $\varepsilon$-constraint method is a multi-objective optimization algorithm [13], also known as the main objective method. Its main idea is to select a main objective function, and transform the rest of the objective functions into constraints, and the multi-objective optimization problem is transformed into a single-objective optimization problem.

### 3.1 Main objective selection

The objectives in a multi-objective optimization problem are usually in conflict with each other and cannot reach the optimum at the same time. In order to make the resulting Pareto solution set more uniformly distributed, one of the most conflicting objectives can be selected as the main objective [14]. In this paper, the Pearson correlation coefficient is used to calculate the relationship between the objective functions, and the value of Pearson correlation

coefficient is in the range of [-1,1], and when its value is closer to -1, it indicates that the negative correlation between the two objective functions is stronger [15], i.e., the conflict is stronger.

First, single-objective optimization is performed for each objective and the corresponding optimal solution $x_n^*$, $n = 1,2,3$. At this time, $x_n^*$ is substituted into the other objective functions, and the payment matrix can be obtained, as shown in Eq (22). Next, the Pearson correlation coefficients between any two objective functions are calculated by Eq (23), and for the three-objective optimization problem, it is obvious that two Pearson correlation coefficients exist for each objective function. Finally, the objective function with the smallest sum of Pearson correlation coefficients is selected as the main objective function, when the main objective function is conflict with the other two objectives.

$$Pay = \begin{pmatrix} f_1(x_1^*) & f_2(x_1^*) & f_3(x_1^*) \\ f_1(x_2^*) & f_2(x_2^*) & f_3(x_2^*) \\ f_1(x_3^*) & f_2(x_3^*) & f_3(x_3^*) \end{pmatrix} \tag{22}$$

$$\rho_{f_i f_j} = \frac{\sum_{n=1}^{3} \left( f_i(x_n^*) - \bar{f}_i \right) \left( f_j(x_n^*) - \bar{f}_j \right)}{\sqrt{\sum_{n=1}^{3} \left( f_i(x_n^*) - \bar{f}_i \right)^2} \sqrt{\sum_{n=1}^{3} \left( f_j(x_n^*) - \bar{f}_j \right)^2}} \tag{23}$$

Where, $\rho_{f_i,f_j}$ denotes the Pearson correlation coefficient of the objective function $f_i$ and $f_j$, and $\bar{f}_i$ and $\bar{f}_j$ are the average values of the different optimal solutions $x_n^*$ corresponding to the objective function $f_i$ and $f_j$, which can be calculated by Eqs (24) and (25), respectively.

$$\bar{f}_i = \frac{1}{3} \sum_{n=1}^{3} f_i(x_n^*) \tag{24}$$

$$\bar{f}_j = \frac{1}{3} \sum_{n=1}^{3} f_j(x_n^*) \tag{25}$$

## 3.2 Improved $\varepsilon$-constraint method

After determining the main objective function, the rest of the objective functions can then be transformed into constraints. For the sake of illustration, $f_1$ is selected as the main objective function. Based on the $\varepsilon$-constraint method, the original model Eq (21) can be transformed into a single objective optimization model as follows:

$$\min f_1(x)$$
$$\text{s.t.} \begin{cases} g(x) = 0 \\ h(x) \leq 0 \\ f_2(x) \leq \varepsilon_2, f_3(x) \leq \varepsilon_3 \end{cases} \tag{26}$$

Where, $\varepsilon_i$ is the upper bound of each objective function $f_i(x)$ except the main objective function. The range of values of $\varepsilon_i$ can be determined by solving min $f_i(x)$ and max $f_i(x)$ to obtain $f_{i\min}$ and $f_{i\max}$. The traditional $\varepsilon$-constraint method divides the range of values of $\varepsilon_i$ into $q$-

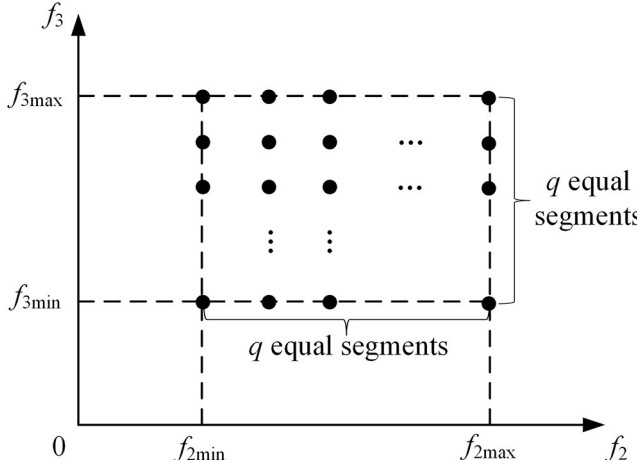

**Fig 1. Grid point of the traditional $\varepsilon$-constraint method.**

equal segments, resulting in uniformly distributed grid points, as shown in Fig 1. Each grid point corresponds to a set of values of $\varepsilon_i$, and solving Eq (26) at different values of $\varepsilon_i$ can obtain the corresponding Pareto solution. By traversing all the uniformly distributed grid points, the set of Pareto solutions can be obtained, from which a solution is selected as the compromise optimal solution of the multi-objective optimization problem.

Obviously, as the value of $q$ increases, the number of the grid points will increase rapidly, and the comprehensive optimization degree of the selected compromise optimal solution is higher. However, the increasing number of the grid points will seriously affect the computation efficiency. To address this problem, this paper proposes an improved $\varepsilon$-constraint method to transform the original model Eq (21) into the two-layer optimization model shown in Eq (27).

$$\min_{\mu_2,\mu_3} d = \sqrt{\bar{f}_1^2(x_1^*) + \bar{f}_2^2(x_1^*) + \bar{f}_3^2(x_1^*)}$$

$$\text{s.t.} \begin{cases} 0 \leq \mu_2 \leq 1, 0 \leq \mu_3 \leq 1 \\ x_1^* = \begin{cases} \arg\min\limits_{x} f_1(x) \\ \text{s.t.} \begin{cases} g(x) = 0 \\ h(x) \leq 0 \\ \varepsilon_2 = f_{2\min} + \mu_2(f_{2\max} - f_{2\min}) \\ \varepsilon_3 = f_{3\min} + \mu_3(f_{3\max} - f_{3\min}) \\ f_2(x) \leq \varepsilon_2, f_3(x) \leq \varepsilon_3 \end{cases} \end{cases} \end{cases} \tag{27}$$

In the inner layer optimization model, auxiliary variables $\mu_2$ and $\mu_3$ are introduced and satisfy $\mu_2, \mu_3 \in [0,1]$, then the values of $\varepsilon_2$ and $\varepsilon_3$ can be uniquely determined by $\mu_2$ and $\mu_3$. Each

time the inner layer optimization model is solved, the Pareto solution corresponding to a certain set of auxiliary variables $(\mu_2, \mu_3)$ is obtained and this solution is transferred to the outer layer optimization model. In the outer layer optimization model, each objective function is normalized to avoid the influence caused by the value range, as shown in Eq (28). Since the three objectives cannot achieve the optimal value at the same time, the Utopia point is an unattainable ideal optimal solution. If the Euclidean distance between a solution and the Utopia point is smaller than other solutions, it can be considered as the highest degree of comprehensive optimal solution. Therefore, the objective function of the outer optimization model is to find a set of $(\mu_2, \mu_3)$ that minimizes the Euclidean distance between the corresponding Pareto solution and the Utopia point $(f_{1\min}, f_{2\min}, f_{3\min})$. In this paper, a simplex search algorithm [16] is used to directly search for the optimal set of $(\mu_2, \mu_3)$ within the range of values of $\mu_2$ and $\mu_3$. By solving the two-layer optimization model, the comprehensive optimal solution can be obtained directly, and it is no longer necessary to traverse every grid points. Therefore, the computation efficiency of the proposed method is significantly improved.

$$\bar{f}_i = \frac{f_i - f_{i\,\min}}{f_{i\,\max} - f_{i\,\min}} \tag{28}$$

The simplex search algorithm was proposed by mathematicians J.A.Nelder and R.Mead, and its solution idea is as follows: let the optimization problem contain n independent variables, firstly, select n+1 points in the feasible solution space to form the initial simplex, then solve the objective function value corresponding to each point of the simplex and sort, and then get the better points in the feasible solution space through reflection and contraction methods to replace the worst points in the initial simplex, and then get the new simplex. The worst point in the feasible solution space is obtained by reflection and contraction, and the new simplex is obtained. By repeating the above process, the simplex is transferred, deformed and shrunk, and gradually approaches the optimal point. The iterative process of the algorithm ends when the difference between the objective function values of the points in the simplex is less than the set value, and the optimal point of the simplex is then taken as the optimal solution.

The flowchart of the algorithm is shown in Fig 2 with the following steps:

**Step 1**: Generate the initial simplex. Considering that the outer optimization model contains two independent variables, $\mu_2$ and $\mu_3$, three points $X_j(\mu_2, \mu_3)$ in the feasible solution space are selected, $j$ = 1,2,3. Substituting and solving the inner optimization model, the Euclidean distances, $d(X_j)$, from the Pareto solution to the Utopia point corresponding to each initial point are obtained, and the initial simplex is generated.

**Step 2**: Convergence judgment of iteration. According to the Euclidean distance $d(X_j)$ to rank, determine the optimal point $X_L$, the next best point $X_G$ and the worst point $X_H$. Calculate the root mean square error of the Euclidean distance corresponding to each point in the simplex, as shown in Eq (29). If the *RMSE* is less than the set accuracy $\tau$, the iteration ends, and the Pareto solution corresponding to the optimal point $X_L$ at this point is taken as the compromise optimal solution. Otherwise, go to **Step 3**.

$$RMSE = \sqrt{\frac{1}{3}\sum_{j=1}^{3}\left(d(X_j) - \frac{1}{3}\sum_{j=1}^{3}d(X_j)\right)^2} \tag{29}$$

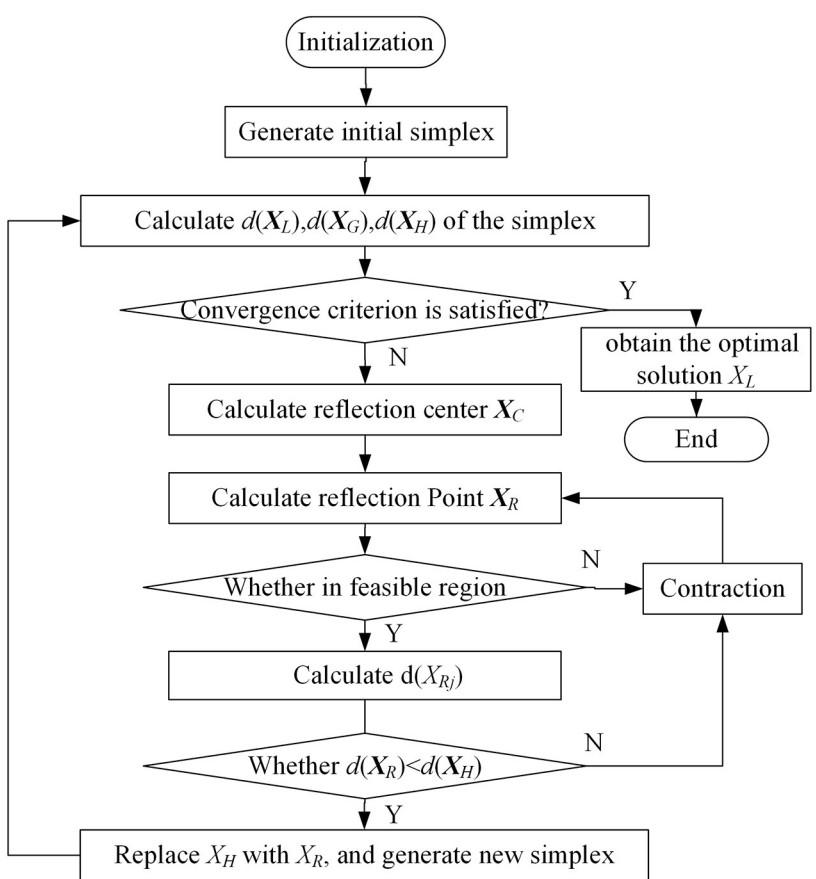

**Fig 2. Flow chart of simplex search algorithm.**

**Step 3**: Calculate the reflection center $X_C$. Calculate the form center of the remaining points in the simplex, except the worst point, as the reflection center $X_C$ according to Eq (30). In general, the direction of the worst point $X_H$ pointing to the reflection center $X_C$ is considered to be the direction in which the objective function is more optimal.

$$X_C = \frac{1}{2}(X_L + X_G) \tag{30}$$

**Step 4**: Calculate the reflection point $X_R$. Calculate the reflection point $X_R$ according to the reflection Eq (31) and check whether it is within the feasible solution space, i.e., whether it satisfies $\mu_2, \mu_3 \in [0,1]$. If it is not in the feasible solution space, recalculate it according to the contraction Eq (32) until a reflection point $X_R$ in the feasible solution space is obtained.

$$X_R = X_C + K_R(X_C - X_H) \tag{31}$$

$$X_R = X_C + K_S(X_R - X_C) \tag{32}$$

**Step 5**: Generate a new simplex. Calculate $d(X_R)$ corresponding to the reflection point $X_R$ and compare $d(X_R)$ with $d(X_H)$ corresponding to the worst point $X_H$. If $d(X_R) > d(X_H)$, return to **Step 4** and calculate the new reflection point according to the contraction Eq (32). If $d(X_R) < d(X_H)$, replace the worst point $X_H$ of the original simplex with the reflection point $X_R$ to form the new simplex and return to **Step 2**.

# 4 Case study

## 4.1 Parameters of the case study

The IEEE-33 bus system is used to demonstrate the effectiveness of the proposed method, as shown in Fig 3. The line parameters are shown in the S1 Table of S1 File [17]. The used computer is a PC with an Intel(R) Core(TM) i7-12700 and 32GB of RAM, and the computing platforms are MATLAB 2018b and Gurobi 9.0.2. Bus 18 and bus 30 are connected to photovoltaic power plant and wind farm, and the maximum active output is 6MW and 8MW, respectively. The predicted value of active output is shown in Fig 4. Bus 24 and bus 30 are connected to the energy storage station, and the capacity is 2.5 MW·h. The distribution network is connected to the main grid from bus 1, and by adjusting the power injected from the main grid to the distribution network, the results of the schedule of orderly power utilization can be analyzed under different power gaps. The active load pattern of each user is taken from eight typical load curves in the S2 Table of S1 File [18]. The participation of each user in orderly power

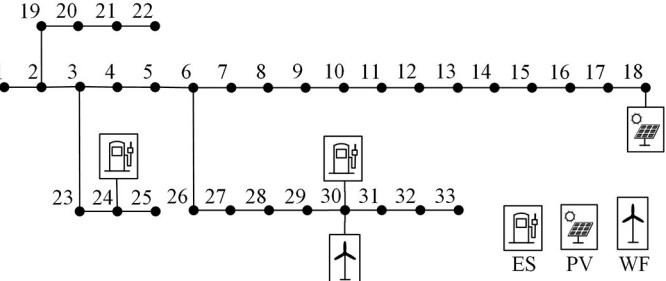

**Fig 3. IEEE-33 bus system.**

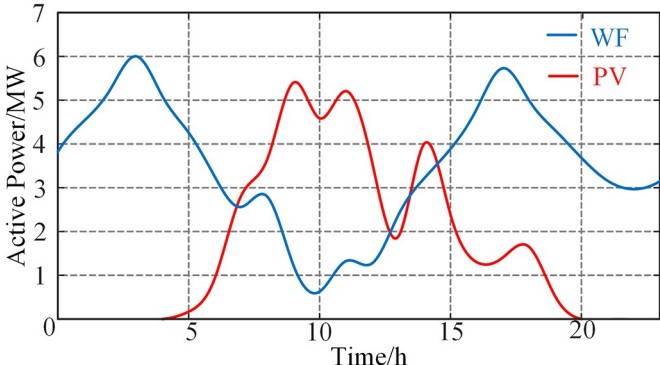

**Fig 4. Forecast active power of wind farm and photovoltaic plant.**

**Table 1. Participation of all users in orderly power utilization.**

| Orderly Power Utilization | Users | User participation |
|---|---|---|
| Peak transferring | All users except 4,15,28 | Maximum time of peak transferring: 2h |
| Peak shifting | 2,3,6,7,10,11,18,20,22,25 | Option number of peak shifting: 3 |
| Peak clipping | All users except 8,13,31 | Maximum peak clipping level:5 |

utilization is shown in Table 1. To ensure the order of the measures of orderly power utilization, the compensation cost coefficient of peak transferring, peak shifting, peak clipping are set as $\lambda_{trans}$ = 300 yuan/h, $\lambda_{shift}$ = 500 yuan/MW·h, $\lambda_{clip}$ = 700 yuan/MW·h. The penalty cost coefficient of renewable energy curtailment is set as $\lambda_{RES}$ = 200 yuan/MW·h. And the charging and discharging operation cost coefficient of the energy storage station is set as $\lambda_{ES}$ = 150 yuan/MW·h.

## 4.2 Multi-objective optimization for orderly power utilization

Considering that there is a 10% power gap in the distribution network at the peak of power utilization, a multi-objective optimization model is carried out. First, single objective optimization is carried out for three objective functions respectively, as shown in Table 2. Calculating the Pearson correlation coefficients between each objective and other objectives, the values of $\rho_{f1,f2}$, $\rho_{f1,f3}$ and $\rho_{f2,f3}$ are -0.94, -0.63 and 0.33, respectively. It can be seen that the Pearson correlation coefficients between $f_1$ and $f_2$, and between $f_1$ and $f_3$ are closer to -1, so $f_1$ is selected as the main objective.

Next, the proposed improved $\varepsilon$-constraint method is used to solve the multi-objective optimization model, and the values of each objective function of the resulting compromised optimal solution are 58.36×10³yuan, 33.62×10⁴yuan, and 12.51×10³MW, respectively. Compared with the results in Table 2, it can be seen that the compromised optimal solution is closer to the single-objective optimal value of each objective, and it is a comprehensively optimized solution with a higher degree of optimization. By solving the proposed multi-objective optimization, an orderly power utilization schedule can be obtained. This schedule simultaneously reduces the total operation cost and the cost for users, which can improve the economic benefits of the system operation. Moreover, it reduces the fluctuation of system load in the distribution network and enhances the operation security of the system.

## 4.3 Comparison of different multi-objective optimization methods

To illustrate the effectiveness of the proposed algorithm, it is compared with the traditional $\varepsilon$-constraint method. In this case, the traditional $\varepsilon$-constraint method is divided into 10 equal parts, 20 equal parts, and 40 equal parts for the range of values of each objective, which corresponds to the need to solve 121 (11 × 11), 441 (21 × 21), and 1681 (41 × 41) grid points, respectively. Solving Eq (26) yields a Pareto solution corresponding to a grid point, and the set of Pareto solutions can be obtained by traversing all the grid points, and then the solution with

**Table 2. Results of single objective optimization.**

| Objective | $f_1$/10³yuan | $f_2$/10⁴ yuan | $f_3$/10³MW |
|---|---|---|---|
| min $f_1$ | 52.13 | 41.35 | 17.42 |
| min $f_2$ | 70.24 | 30.72 | 13.58 |
| min $f_3$ | 63.85 | 37.68 | 9.64 |

**Table 3. Comparison of multi-objective optimization results.**

| Method | $f_1/10^3$yuan | $f_2/10^4$yuan | $f_3/10^3$MW | d | CPU time/s |
|---|---|---|---|---|---|
| $\varepsilon$-constraint(10) | 62.07 | 37.13 | 13.85 | 0.554 | 311.2 |
| $\varepsilon$-constraint(20) | 61.74 | 35.41 | 13.42 | 0.481 | 1084.6 |
| $\varepsilon$-constraint(40) | 60.18 | 34.78 | 12.64 | 0.427 | 3948.5 |
| improved $\varepsilon$-constraint | 58.36 | 33.62 | 12.51 | 0.415 | 81.7 |

the smallest distance from the utopia point is selected from the solution set as the compromise optimal solution. The computational results are shown in Table 3.

It can be seen that for the traditional $\varepsilon$-constraint method, the number of solutions in the Pareto solution set increases significantly as the number of equal parts increases. Although the obtained compromise optimal solution has higher degree of comprehensive optimization, the computation time also increases significantly. Compared with the traditional $\varepsilon$-constraint method, the CPU time of the improved $\varepsilon$-constraint method is smaller. Based on several initial points, the proposed method generates the simplex in the feasible solution space and obtain the compromise optimal solution, which avoids solving a large number of Pareto solutions corresponding to all the mesh points. Therefore, the computation efficiency can be significantly improved. In addition, compared with the traditional $\varepsilon$-constraint method, the value of each objective function is better and the distance $d$ between the compromise optimal solution and the Utopia point is also smaller. This is because the direct search algorithm takes points continuously in the feasible solution space, while the traditional algorithm takes points discretely in a grid, which is easy to miss the more optimized solutions existing between the discrete points. In summary, the proposed method significantly reduces computation time while obtaining a better compromise optimal solution.

## 4.4 Analysis of orderly power utilization under different power shortages

In order to analyze the schedule of orderly power utilization under different power shortages, the proposed model are carried out by considering the existence of 5% and 15% power shortages in the peak period, as shown in Table 4.

It can be seen that when the power shortage is only 5%, all users of the system do not need to participate in peak clipping, but only through peak transferring and peak shifting. When the power shortage expands to 10%, the load adjustment caused by all kinds of orderly power utilization measures increases. Among them, the adjustment amount of peak shifting is larger than the adjustment amount of peak transferring. This is because in response to the power shortage, peak transferring is the first orderly use of electricity measures taken, and as the shortage increases, the space for users to adjust the peaking has been very small, so the adjustment amount of peak shifting is more varied. When the power shortage continues to expand to 15%, more users need to participate in peak clipping to achieve active power balance.

**Table 4. Results of orderly power utilization under different power shortages.**

| Power shortage | Peak transferring/MWh | Peak shifting/MWh | Peak clipping/MWh |
|---|---|---|---|
| 5% | 27.3 | 13.4 | 0 |
| 10% | 33.5 | 29.7 | 8.7 |
| 15% | 34.8 | 37.1 | 28.6 |

## 5 Conclusion

This paper establishes a multi-objective optimization model for orderly power utilization in active distribution networks. Convex relaxation techniques are adopted to transform the original model into a MISOCP model, and the improved $\varepsilon$-constraint method is proposed to solve the model efficiently. Compared with the traditional single objective optimization model, the solution of multi-objective optimization is a solution with a high degree of comprehensive optimization. Compared with the traditional $\varepsilon$-constraint method, the objective value of the proposed method has been improved by 15.9% and the computation time is significantly reduced by 92.4% in the case studies.

## Supporting information

**S1 File.**
(XLSX)

## Author Contributions

**Conceptualization:** Xin Wen.

**Data curation:** Hui Li, Xiaoqiang Wu.

**Formal analysis:** Xin Wen, Hui Li, Liu Siliang.

**Investigation:** Liu Siliang.

**Methodology:** Xin Wen, Xiaoqiang Wu, Liu Siliang, Guohua Huang.

**Resources:** Hui Li, Liu Siliang.

**Software:** Xin Wen, Hui Li.

**Supervision:** Hui Li.

**Visualization:** Guohua Huang.

**Writing – original draft:** Yiwei Li, Guohua Huang.

**Writing – review & editing:** Yiwei Li, Guohua Huang.

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
