## [Decision Letter · Decision Letter 0]

30 Jan 2024

PONE-D-23-41665Multi-objective Optimal Decision for Orderly Power Utilization Based on Improved ε-Constraint MethodPLOS ONE

Dear Dr. huang,

Thank you for submitting your manuscript to PLOS ONE. After careful consideration, we feel that it has merit but does not fully meet PLOS ONE’s publication criteria as it currently stands. Therefore, we invite you to submit a revised version of the manuscript that addresses the points raised during the review process.

In particular, different final recommendations have been received by three expert reviewers. Apart from specific minor comments, it appears that the scientific contribution of the manuscript, and its novelty with respect to the existing state of the art, remains as the main shortcoming of the manuscript. If the Authors believe that they are able to address this last concern, then they are invited to resubmit a revised version of the manuscript. Please note that the revised manuscript would be evaluated mostly in terms of its original scientific contribution.

We look forward to receiving your revised manuscript.

Kind regards,

Emanuele Crisostomi, PhD

Academic Editor

PLOS ONE

Journal Requirements:

2. Thank you for submitting the above manuscript to PLOS ONE. During our internal evaluation of the manuscript, we found significant text overlap between your submission and previous work in the [introduction, conclusion, etc.].

Please revise the manuscript to rephrase the duplicated text, cite your sources, and provide details as to how the current manuscript advances on previous work. Please note that further consideration is dependent on the submission of a manuscript that addresses these concerns about the overlap in text with published work.

[If the overlap is with the authors’ own works: Moreover, upon submission, authors must confirm that the manuscript, or any related manuscript, is not currently under consideration or accepted elsewhere. If related work has been submitted to PLOS ONE or elsewhere, authors must include a copy with the submitted article. Reviewers will be asked to comment on the overlap between related submissions (http://journals.plos.org/plosone/s/submission-guidelines#loc-related-manuscripts).]

We will carefully review your manuscript upon resubmission and further consideration of the manuscript is dependent on the text overlap being addressed in full. Please ensure that your revision is thorough as failure to address the concerns to our satisfaction may result in your submission not being considered further.

"This research received funding Science and Technology Project of China Southern Power Grid Corporation (GZHKJXM20210056 (080036KK52210003))."

"This research received funding Science and Technology Project of China Southern Power Grid Corporation (GZHKJXM20210056 (080036KK52210003))."

"This research received funding Science and Technology Project of China Southern Power Grid Corporation (GZHKJXM20210056 (080036KK52210003))."

7. In the online submission form, you indicated that [he data used to support the study are available upon request to the corresponding author.]. 

8. PLOS requires an ORCID iD for the corresponding author in Editorial Manager on papers submitted after December 6th, 2016. Please ensure that you have an ORCID iD and that it is validated in Editorial Manager. To do this, go to ‘Update my Information’ (in the upper left-hand corner of the main menu), and click on the Fetch/Validate link next to the ORCID field. This will take you to the ORCID site and allow you to create a new iD or authenticate a pre-existing iD in Editorial Manager. Please see the following video for instructions on linking an ORCID iD to your Editorial Manager account: https://www.youtube.com/watch?v=_xcclfuvtxQ

9. We note you have included a table to which you do not refer in the text of your manuscript. Please ensure that you refer to Table 3 in your text; if accepted, production will need this reference to link the reader to the Table.

Reviewers' comments:

Reviewer's Responses to Questions

**Comments to the Author**

1. Is the manuscript technically sound, and do the data support the conclusions?

Reviewer #1: No

Reviewer #2: Yes

Reviewer #3: Yes

2. Has the statistical analysis been performed appropriately and rigorously? 

Reviewer #1: No

Reviewer #2: Yes

Reviewer #3: N/A

3. Have the authors made all data underlying the findings in their manuscript fully available?

Reviewer #1: Yes

Reviewer #2: No

Reviewer #3: Yes

4. Is the manuscript presented in an intelligible fashion and written in standard English?

Reviewer #1: No

Reviewer #2: Yes

Reviewer #3: Yes

5. Review Comments to the Author

Reviewer #1: 1. English should be improved.

2. Title: Please use the literature background to develop a better title in order to catch the prospective reader's attention.

3. Abstract: It should be rewritten in terms of aim, background, motivation, and significant results. Your abstract should clearly state the essence of the problem you are addressing, what you did, and what you found and recommended. That would help prospective readers of the abstract decide if they wish to read the entire article.

4. The main concern I have about the paper is with respect to the contributions of your work. The methodology used has few outstanding innovation points. So what innovative work have you done compared with previous studies?

5. How does your work contribute to this field? It is not clearly stated in the abstract and conclusion sections.

6. Introduction: it is not well organized. With the current introduction, it is not possible to understand the gap between the previous research and the novelty of the current works. The introduction should be fully reorganized to show the difference between the current work and the previous one.

7. What is the new devolved methodology? It is not clearly presented.

8. The provided results are limited, not convincing, and insufficient to publish in this quality journal.

9. Discussion of the results should provide useful insights. The results should be further elaborated to display how they could be utilized in real-world situations. The authors should further develop critical assessment in the results discussion.

10. Conclusions: It is poorly written and lacks an evaluation of the significant and main findings from the current work that have been found.

Reviewer #2: Authors proposed a multi-objective optimization to reduce the costs of electricity supply, including total operation costs, costs for the users, and minimization of system load fluctuations. Three demand response programs, such as peak transferring, peak shifting and peak clipping were considered. The work is interesting and relevant, however, some details should be clarified before publication.

I would suggest extending and diversifying the literature review. A part of the paper contributions is the improved ε-constraint method, as there are several methods proposed based on the original method, it could be relevant to include them in the literature review and comment the advantages/suitability to the case study of the proposed algorithm.

Definitions of peak transferring and peak shifting looked very similar to me. If I did not misunderstand something, in both cases the loads are shifted in time, but the final total consumption does not change, and it looks like peak transferring is a particular case of peak shifting. If this is the case, could these two values be introduced as one? If not, could you better clarify the difference between the two elements?

The choices of case study parameters (especially costs) should be stated and justified, for example, it is important to specify the discount rates applied to each element (peak transferring, peak shifting and peak clipping). Also, the term “charging and discharging operation cost of the energy storage plant” is not clear to me. Why operation costs of batteries depend on the charging and discharging power at the time period?

The whole text should be proofread to solve some typos. For example, confused words (“peak shifting, peak shifting, and peak clipping,…”, “staggering compensation coefficient”), unclear sentences structure (“in the feasible domain are selected. substituting…”), missing verbs in sentences. Variables in the equations and in the text must correspond (for example, utrans i,h, QLi,t). Also, location of legend in the Figure is confusing, it looks like ES, PV and WF are located at bus 33.

Reviewer #3: This paper presents an approach to the optimal orderly power utilization problem using an enhanced epsilon-constraint optimization method.

The authors should take into consideration the following issues:

- Page 2: sentence “Since power cut will seriously national production and life …” must be rephrased!

- Page 2: pay attention to repetitions: “the graded peak shifting and graded peak shifting”.

- Page 2, lines before eq. (1): again, pay attention to repetitions – “for peak shifting, peak shifting, and peak clipping”.

- Page 3 and forward: when introducing different “lambda” coefficients, you should define their meaning, including units of measurement, e.g. lambda_trans, lambda_shift, lambda_clip, lambda_RES or lambda_ES. Basically, they should represents some kind of prices, but you should clearly define them.

- Page 3 and forward: when use in text, different variables should use sub- and super-scripts, e.g. “utrans i,h” but other similar notations exist.

- Page 3: as eq. (5) suggests, C_RES,t can be positive or negative, depending on the sign of differences between forecasted and scheduled output of photovoltaic and wind farms. Please, comment on this subject!

- Page 9, Step 5 of the algorithm: you refer to “the contraction formula (30)”. Please, check if the reference to eq. (30) is correct!

- Page 11, title of subsection D: explain what do you understand by “power gap”.

6. PLOS authors have the option to publish the peer review history of their article (what does this mean?). If published, this will include your full peer review and any attached files.

Reviewer #1: No

Reviewer #2: No

Reviewer #3: No

---

## [Author Response · Author response to Decision Letter 0]

20 Jun 2024

The file "Response to Reviewers" has been submitted via email. Thank you!

---

## [Decision Letter · Decision Letter 1]

13 Aug 2024

Multi-objective Optimal Decision for Orderly Power Utilization Based on Improved ε-Constraint Method in Active Distribution Networks

PONE-D-23-41665R1

Dear Dr. huang,

We’re pleased to inform you that your manuscript has been judged scientifically suitable for publication and will be formally accepted for publication once it meets all outstanding technical requirements.

Kind regards,

Emanuele Crisostomi, PhD

Academic Editor

PLOS ONE

Additional Editor Comments (optional):

Reviewers' comments:

Reviewer's Responses to Questions

**Comments to the Author**

1. If the authors have adequately addressed your comments raised in a previous round of review and you feel that this manuscript is now acceptable for publication, you may indicate that here to bypass the “Comments to the Author” section, enter your conflict of interest statement in the “Confidential to Editor” section, and submit your "Accept" recommendation.

Reviewer #1: (No Response)

Reviewer #3: All comments have been addressed

2. Is the manuscript technically sound, and do the data support the conclusions?

Reviewer #1: Partly

Reviewer #3: Yes

3. Has the statistical analysis been performed appropriately and rigorously? 

Reviewer #1: No

Reviewer #3: N/A

4. Have the authors made all data underlying the findings in their manuscript fully available?

Reviewer #1: No

Reviewer #3: Yes

5. Is the manuscript presented in an intelligible fashion and written in standard English?

Reviewer #1: No

Reviewer #3: Yes

6. Review Comments to the Author

Reviewer #1: There is no novelty in the proposed method.

The introduction is very poor.

The proposed method is unclear.

More descriptions should be added.

Reviewer #3: No more comments. The authors have positively responded to all comments, suggestions or questions raised by the reviewer.

7. PLOS authors have the option to publish the peer review history of their article (what does this mean?). If published, this will include your full peer review and any attached files.

Reviewer #1: No

Reviewer #3: No

---

## [Editor Report · Acceptance letter]

14 Oct 2024

PONE-D-23-41665R1 

PLOS ONE

Dear Dr. Huang, 

I'm pleased to inform you that your manuscript has been deemed suitable for publication in PLOS ONE. Congratulations! Your manuscript is now being handed over to our production team.

Kind regards, 

on behalf of

Professor Emanuele Crisostomi 

Academic Editor

PLOS ONE